# Analysis of the Function of LncRNA-*MSTRG.16919.1* in BHV-1-Infected Bovine Kidney Subculture Cells by Transcriptome Sequencing

**DOI:** 10.3390/v14102104

**Published:** 2022-09-22

**Authors:** Fan Zhang, Kunsheng Jiang, Yuchun Wang, Jinzhu Ma, Baifen Song

**Affiliations:** 1The College of Veterinary Medicine, China Agricultural University, Beijing 100091, China; 2The College of Life Science and Technology, Heilongjiang Bayi Agricultural University, Daqing 163319, China

**Keywords:** BHV−1, bioinformatic analysis, lncRNA, MDBK, transcriptome sequencing

## Abstract

Infection of cattle with bovine herpesvirus type 1 (BHV−1) can lead to upper respiratory tract disease, conjunctivitis, or genital disease and cause serious economic losses to the cattle industry worldwide. The role of long noncoding RNAs in BHV−1 infection is not well understood. To explore the role of lncRNA−*MSTRG.16919.1* in bovine herpes virus type I (BHV−1) infected MDBK cells, the lncRNA−*MSTRG.16919.1* gene was silenced and sequenced transcriptome and sequencing data were analyzed by Edge R software, Gene Ontology (GO), the Kyoto Encyclopedia of Genes and Genomes (KEGG), and an interaction network of proteins. Real−time quantitative PCR (RT−qPCR) and Western blotting were used to verify the results of bioinformatic analyses. The results showed that 1151 differential genes were obtained in the siRNA−*MSTRG.16919.1* group compared with an NC group. Compared with BHV−33 h, 6586 differentially expressed genes were obtained. A total of 498 differentially expressed genes were screened from the two groups. To verify the accuracy of the sequencing, six genes were randomly selected for RT−qPCR, and the results showed that the expression trend of selected genes was consistent with the sequencing results. GO enrichment analysis showed that the differential genes were related to such biological processes as nucleotide binding, enzyme binding, cell cycle, and glial macromolecule metabolism. KEGG analysis enriched 378 and 2634 signaling pathways, respectively, that were associated with virus infection, ubiquitin−mediated protein hydrolysis, phosphoinositol metabolism, apoptosis, and other metabolic pathways. The STRING protein interaction database was used to analyze the interaction network of proteins encoded by differential genes, and the degree algorithm in Cytoscape was used to screen the top 20 proteins. The results showed that SKIV2L2, JAK2, PIK3CB, and MAPK8 were related to virus infection. Western blot analysis of TNF, NF−κB, MAPK8, MAPK9, and MAPK10 proteins showed that lncRNA−*MSTRG.16919.1* was involved in regulating the expression of these functional proteins. The results of this study provide basic information for exploring the function and regulatory mechanism of lncRNA−*MSTRG.16919.1* in organisms and help for further studying the interaction between virus and host.

## 1. Introduction

Infectious bovine rhinotracheitis (IBR), also known as necrotizing rhinitis and red nose disease, is a respiratory contact infection of cattle caused by bovine herpesvirus type 1 (BHV−1). Its clinical manifestations are varied, including respiratory tract changes, conjunctivitis, abortion, mastitis, and induced calf encephalitis. This disease is currently prevalent in the world and has a significant impact on milk yield of dairy cattle, fecundity of bulls, and servitude of draft cattle [1,2]. It belongs to the family Herpesviridae and subfamily Alphaherpesvirinae. The members of this family are large, enveloped, and double−stranded DNA viruses, and among them, BHV−1 is an important member that causes severe economic losses to the cattle industry worldwide [3,4].

Long noncoding RNA (lncRNA) is a class of noncoding RNA with a length of more than 200 nt, transcribed by RNA polymerase II or III, existing in the nucleus and cytoplasm, with long transcripts and lack of open reading frame, and unable to translate proteins in cells [5]. They participate in gene regulation in the form of RNA in epigenetic regulation, transcriptional regulation, and posttranscriptional regulation [6]. Studies have shown that lncRNA is closely related to tissue differentiation, organogenesis, and tumorigenesis [7]. In addition, lncRNA is also closely related to virus infection, and it has been found that many viruses, such as human immunodeficiency virus (HIV), human hepatitis B virus (HBV), human hepatitis C virus (HCV), and avian bursa of bursa virus (IBDV) can induce cells to produce lncRNAs. This lncRNA production plays a key role in viral replication, interferon production, NF−κB, STAT, and other pathways [8,9,10,11], but it has not been reported whether similar lncRNA is produced after BHV−1 infection.

To understand the mechanism of lncRNA in BHV−1 infection, bovine kidney subculture (MDBK) cells were infected with BHV−1 in this study. Transcriptome sequencing in separate phases was conducted to obtain the lncRNA expression profile of BHV−1−infected MDBK cells [12], a novel and upregulated lncRNA−*MSTRG.16919.1* was screened out, but the function of lncRNA−*MSTRG.16919.1* is unknown, so transcriptome sequencing was performed to understand the function of the lncRNA−*MSTRG.16919.1* gene after it was silenced.

## 2. Materials and Methods

### 2.1. Experimental Materials

Bovine herpesvirus type 1 (BHV−1) and MDBK were stored in the cell biology laboratory of Heilongjiang Bayi Agricultural University (Daqing, China). The Ribo FECTTMCP transfection reagent and Ribo lncRNA smart silencer interfering plasmid were designed and synthesized by Guangzhou RiboBio. RNA extraction, cDNA reverse transcription, and fluorescence quantitative PCR kits were all purchased from Nanjing Vazyme Biotechnology. A BCA protein concentration determination kit and RIPA lysate were purchased from Solarbio. Beta actin and sheep anti−mouse IgG−HRP antibodies were purchased from Bioss. NF−κB, JNK and IκB antibodies were purchased from Affinity Biosciences.

### 2.2. Sample Preparation

The MDBK cells were cultured in Dulbecco’s modified Eagle’s medium (DMEM) containing 10% fetal bovine serum (Biological Industries, Kibbutz Beit−Haemek, Israel) and cultured in a 5% CO_2_ incubator at 37 °C. On the day before transfection, 1 × 10^6^ cells were inoculated in a 6−well plate culture well containing a proper amount of complete medium consisting of 10% FBS with 1% penicillin and streptomycin dual antibiotic, so that the cell density during transfection reached 80%. The experimental groups were as follows. The siRNA−*MSTRG.16919.1* group was the experimental group, in which MDBK cells were infected with BHV−1 for 33 h and silenced lncRNA−*MSTRG.16919.1*; siRNA−*MSTRG.16919.1 NC* group was MDBK cells infected with BHV−1 for 33 h and silenced an unrelated sequence was the negative control group of the siRNA−*MSTRG.16919.1* group; the BHV−1 33 h group was the MDBK cells infected with BHV−1 33 h as the reference group; and the blank group was the MDBK cell group without any treatment. BHV−1 was inoculated into monolayer MDBK cells at MOI (multiplicity of infection) = 1 dose and then transfected after 1 h.

RiboFECT™ CP buffer (10×) was diluted with sterile phosphate buffer saline (PBS) to prepare RiboFECT CP Buffer (1×). RiboFECT CP reagent was removed from 4 °C storage and fully oscillated in a vortex oscillator at room temperature. A 20 μL 20 μM smart silencer siRNA storage solution was diluted with 120 μL 1× riboFECT CP buffer, mixed gently, and incubated at room temperature for 5 min. Next, 12 μL riboFECT CP reagent was added, gently blown, and mixed and incubated at room temperature for up to 15 min. After incubation, the RiboFECT CP mixture was added to 1848 μL cell culture medium, gently mixed, and the culture plate cultured in a carbon dioxide incubator at 37 °C. After 18 h of infection, the occurrence of cytopathy (CPE) was observed and photos taken every hour. When cracks were observed in the cell layer, the samples were collected and washed with Hank’s solution, then digested with Trizol. Finally, the samples were collected into a nuclease−free cell cryopreservation tube, sealed with a sealing membrane, and frozen in liquid nitrogen. One part of the samples was used to verify the gene silencing effect. After lncRNA−*MSTRG.16919.1* silencing was confirmed, the other part of the samples was transported on dry ice to Guangzhou Gidio Biotechnology for high−throughput sequencing.

### 2.3. RNA−seq

Total RNA was extracted from siRNA*−MSTRG.16919.1* samples, siRNA−*MSTRG.16919.1NC* negative control samples, and BHV−33 h samples. The RNA quality was evaluated on an Agilent 2100 Bioanalyzer (Agilent Technologies, Palo Alto, CA, USA) and examined using agarose gel electrophoresis without RNase. After extracting total RNA, eukaryotic mRNA was enriched with oligomer (DT) beads. The enriched mRNA was cut into short fragments with fragment buffer and reverse−transcribed into cDNA with NEBNext Ultra RNA Library Prep Kit for Illumina (NEB#7530, New England Biolabs, Ipswich, MA, USA). The purified double−stranded cDNA fragments were end repaired, A base added, and ligated to Illumina sequencing adapters. The ligation reaction was purified with the AMPure XP Beads (1.0X). Ligated fragments were subjected to size selection by agarose gel electrophoresis and polymerase chain reaction (PCR) amplified. The resulting cDNA library was sequenced using Illumina Novaseq6000 by Gene Denovo Biotechnology (Guangzhou, China). Reads obtained from the sequencing machines included raw reads containing adapters or low−quality bases that would affect the following assembly and analysis. Thus, to get high−quality clean reads, reads were further filtered by fastp (version 0.18.0). The steps for read filtering involved removing reads with adapters, reads containing more than 10% N, reads that were all A bases and of low quality, where the base number of mass value Q ≤ 20 accounted for more than 50% of the whole read.

### 2.4. Statistics on the Number of Differentially Expressed mRNA

The bioinformatic software SIMCA 14.1 (Umetrics, Sverige) was used to analyze the sequencing data and differentially expressed mRNA. The screening criteria of differentially expressed mRNA were |log_2_(FC)| > 1 and FDR < 0.05. Principal component analysis (PCA) was used to analyze and calculate the Pearson’s correlation coefficient between samples, to understand the repeatability between samples and assist in excluding outliers. According to significantly different mRNA in each comparison group, volcanic map analysis was conducted to visually show the different mRNA among the comparison groups. mRNA with similar expression patterns may have common functions or participate in common metabolic pathways and signal pathways, so the analysis was mainly divided into three parts [13]: normalize the read count, calculate the probability of hypothesis test (*p* value) according to the model, and carry out multiple hypothesis tests and corrections to obtain the error−detection rate (FDR). The read count was normalized, the probability of hypothesis test (*p* value) calculated according to the model. and then multiple hypothesis tests and corrections conducted to obtain FDR values or error−detection rates.

### 2.5. Differentially Expressed mRNA by Real−Time Quantitative PCR (RT−qPCR)

According to the sequencing results, three differential genes were randomly selected for upregulation and downregulation. The primers used in RT−qPCR were designed by Primer Premier 5 (Premier, British Columbia, BC, Canada) software and compared by NCBI (National Institutes of Health, Bethesda, MD, USA). All primers were synthesized by Shanghai Sangon Bioengineering Technology Service (Table 1). Trizol (Ambion, Austin, TX, USA) reagent was used to extract total RNA from cells and the concentration and quality of extracted RNA were determined by NanoDrop 2000 (Thermo, Waltham, MA, USA) A reverse−transcription kit (Vazyme, Jiangsu, China) was used to synthesize cDNA strands, and cDNA was used as template to detect differentially expressed genes by RT−qPCR to verify the accuracy of sequencing. Ubiquitin C−terminal hydrolase L5 [14] was selected as the internal reference gene and the *t*−test and 2−ΔΔCt method were used to analyze the data of the relative transcription level of each gene. Three multiple holes were set for each gene in each experiment and three independent repeated experiments were conducted. Finally, the average of relative transcription level was calculated. The relative transcription level of each gene was calculated by the following formula: mRNA relative transcription expression = 2−ΔΔCT, ΔCt value = target gene Ct value−internal reference gene Ct value. The results of RT−qPCR were analyzed by GraphPad. Prism, with * *p* < 0.05, ** *p* < 0.01, *** *p* < 0.001, **** *p* < 0.0001.

### 2.6. GO Enrichment Analysis of Target Gene

The differential genes were mapped to each term of the GO database (GO database. Available online: http://www.geneontology.org/ (accessed on 12 June 2022)) [15] and the number of differential genes in each term calculated to get the statistics of the number of differential genes in the list of differential genes with a certain GO function. The hypergeometric test was then used to find out the GO items that were significantly enriched in the differential genes compared with the background [16]. After the calculated *p* value was corrected by Bonferroni, the threshold value was corrected with *p* values less than 0.05 and the GO term satisfying this condition was defined as a GO term that was significantly enriched in differentially expressed proteins. The main biological functions of differentially expressed proteins were determined by GO functional significance enrichment analysis [17].

### 2.7. Kyoto Encyclopedia of Genes and Genomes (KEGG) Enrichment Analysis of Target Genes

Pathway significant enrichment analysis is based on KEGG Pathway as a unit and a hypergeometric test is used to find the pathways that are significantly enriched in differential genes compared with the whole background. Through pathway significance enrichment, the most important biochemical metabolic pathways and signal transduction pathways involved in differential genes can be determined. The calculated *p* value was corrected by FDR and FDR less than 0.05 was taken as the threshold. Pathways satisfying this condition were defined as significantly abundant pathways in differentially expressed genes (DEGs).

### 2.8. Western Blot Verification of Partial Signal Pathways of KEGG Enrichment

According to the signal pathways obtained from KEGG enrichment analysis, some signal pathways related to virus infection were verified by Western blots. The sample was lysed with high−efficiency RIPA cell lysate (Solarbio, Beijing, China). The sample was placed on ice for lysis for 30 min and tapped gently every 10 min. The cells were treated with a cell scraper and transferred to a centrifuge tube and centrifuged for 5 min at 4 °C at 12,000 rpm/min and the supernatant was subpackaged and stored. The protein was quantified by a BCA protein quantitative kit (Solarbio, Beijing, China) and its concentration determined.

PAGE separation gel and concentrated gel were prepared according to the polyacrylamide gel formula (Solarbio, Beijing, China). After electrophoresis at 80 V for 30 min, the voltage was switched to 120 V and then electrophoresis conducted for 1 h. After electrophoresis, the concentrated gel on the gel was removed. The filter paper, nitrocellulose membrane (NC), gel and filter paper were soaked in pre−membrane transfer solution and sequentially placed in a wet transfer tank and bubbles in the filter paper rolled out by a glass rod. After installation, the transfer slot was covered, the power supply turned on at 200 mA constant current and the film rotated for 2 h in an ice bath. After tris buffered saline tween (TBST) cleaning, the transformed NC membrane was soaked in 5% skimmed milk and sealed at room temperature for 2 h. Skimmed milk was removed, and the NC membrane was washed with TBST three times, then 1:1000 diluted primary antibody was added and incubated overnight at 4 °C. The NC membrane was cleaned again with TBST three times, 1:5000 diluted goat anti−rabbit IgG−HRP antibody was added and incubated at room temperature for 1 h. The NC film was then cleaned with TBST three times, then ECL (Merck KGaA, Darmstadt, Germany) chromogenic solution was used to take pictures with an AI600 imaging system and ImageJ image analysis software was used to quantitatively analyze protein.

### 2.9. Protein Interaction Network Analysis

The interaction relationships in the STRING [18] protein interaction database (STRING. Available online: http://string−db.org/ (accessed on 12 June 2022)) [19] were used to analyze the differential gene−protein interaction network. Aiming at the species contained in the database, the differential gene set was extracted from the database and Cytoscape [20] was used to construct the interaction relationship network diagram. For the species not included in the database, the sequence of the target gene set was first aligned with the protein sequence of the reference species contained in the string database by blastx and the interaction network constructed by using the protein interaction relationship of the reference species in the alignment.

## 3. Results

### 3.1. Transcript−Level Detection of Silencing Effect of lncRNA−MSTRG.16919.1

The prepared RNA samples were extracted from total cell RNA, and the expression of lncRNA−*MSTRG.16919.1* was detected by RT−qPCR. The results are shown in Figure 1. It can be seen from the figure that compared with the control group, siRNA−*MSTRG.16919.1* group and BHV−1 33 h expression was significantly downregulated. The silencing efficiency was 86.96%. We believed that lncRNA−*MSTRG.16919.1* was successfully silenced, and follow−up experiments could be carried out.

### 3.2. Analysis of Transcriptome Sequencing Data Quality

To ensure the data quality, the original data should be filtered before information analysis, to reduce the analysis interference brought by invalid data. From Table 2, after data filtering, the number of high−quality reads and the percentage based on raw reads of the three groups of samples were above 99%, which met the needs of subsequent experiments.

### 3.3. Overall Statistical Results of Differential Genes

Transcriptome sequencing was performed on siRNA−*MSTRG.16919.1* samples, siRNA−*MSTRG.16919.1NC* negative control samples, and BHV−33 h samples. According to the comparison results of HISAT2, transcripts were reconstructed by Stringtie and the expression levels of all genes were calculated in each sample by RSEM [21]. The input data of gene differential expression analysis was the read count data obtained from gene expression level analysis, which was analyzed by EdgeR [22] software.

Based on the results of the difference analysis, the genes were screened with FDR less than 0.05 and |log2FC| greater than one as significant difference genes. The results showed that 1151 differential genes were obtained in the siRNA−*MSTRG.16919.1* group compared with the negative control group and 854 genes were upregulated and 297 genes were downregulated (Figure 2a). Compared with BHV−33 h, 6586 differentially expressed genes were obtained, 4127 genes were upregulated and 2459 genes were downregulated, as shown in Figure 2b. A total of 498 duplicate genes were obtained by screening the differential genes of the two groups (Figure 2c).

According to the significantly different genes in each group, volcanic map analysis was conducted. Volcanic mapping visually shows the difference of genes between comparison groups, and the closer the genes at both ends in the map, the greater the difference. The abscissa represents the logarithmic value of the difference multiple between the two groups, and the ordinate represents the negative log10 value of FDR of the difference between the two groups. The results (Figure 3) are consistent with those shown in Figure 2.

### 3.4. Verification of Differentially Expressed mRNA by RT−qPCR

According to the results of RNA−seq, the difference multiple exceeding two and *p* value less than 0.05 were selected as the screening condition, and three upregulated and three downregulated genes were randomly selected from 498 differentially expressed repeat genes for RT−qPCR verification analysis. The upregulated genes were *Cyclin−K, X4, ATP6VOE1* and downregulated genes were *ADGB*, *KCNIP2*, *WDY*. The results showed that the changes in differentially expressed genes were basically consistent with the analysis results of sequencing data, indicating that the RNA−seq data had high reliability and accuracy and could be used for subsequent experimental research, as shown in Figure 4.

### 3.5. GO Enrichment Analysis Results

To further understand the information of differential expression genes, GO enrichment analysis was conducted and it was found that the differential genes of siRNA−*MSTRG.16919.1* group and siRNA−*MSTRG.16919.1NC* group were enriched by GO and annotated by functional classification, among which 619 genes were annotated to molecular function, 620 genes were annotated to cellular component and 666 genes were annotated to biological process. In the siRNA−*MSTRG.16919.1* group and BHV−33 h group, 3952 GO terms were enriched in molecular function, 4280 GO terms in cellular component and 4430 GO terms in biological process. In Top20, lncRNA−*MSTRG.16919.1* was enriched in molecular functions including nucleotide binding, nucleoside triphosphatase activity, purine nucleoside binding and enzyme binding, which indicated that lncRNA−*MSTRG.16919.1* may participate in the binding of related enzymes in cells. The cell components involved cells and cell components, including microtubule cytoskeleton, intracellular part, microtubule tissue center, cytoskeleton part, organelles and spindle microtubules, which showed that lncRNA−*MSTRG.16919.1* affected the gene expression of cell components in MDBK cells. Biological processes included cell cycle process, glial macromolecule metabolism process, and cell metabolism process, suggesting that lncRNA−*MSTRG.16919.1* may affect many biological processes in MDBK cells, as shown in Figure 5 and Figure 6.

### 3.6. KEGG Enrichment Analysis Results

In organisms, different genes coordinate their biological functions, and pathway−based analysis is helpful to further understand the biological functions of genes. KEGG is the primary public database on pathways. KEGG enrichment analysis of siRNA−*MSTRG.16919.1* group and siRNA−*MSTRG.16919.1NC* group, siRNA−*MSTRG.16919.1* group and BHV−33 h group showed that siRNA−*MSTRG.16919.1* group and siRNA−*MSTRG.16919.1NC* group were enriched to 378 signal pathways. The top 20 signaling pathways involved cancer including viral carcinogenesis, colorectal cancer, small cell lung cancer, microRNA in cancer tissue, gastric cancer, endometrial cancer and other cancer pathways. They also included viral infection and included Toll and Imd signaling pathway, TNF signaling pathway, NF−κB signaling pathway, Barr virus infection and MAPK signaling pathway. It also included hepatitis B, ubiquitin−mediated protein hydrolysis, phosphate metabolism, cell cycle, apoptosis and other metabolic pathways.

There were 2634 signal pathways in siRNA−*MSTRG.16919.1* group and BHV−33 h group. The top 20 signaling pathways involved diseases such as fatty liver, non−trembling thermogenesis, Huntington’s disease, Parkinson’s disease, Alzheimer’s disease, chronic myeloid leukemia, intestinal cancer and platinum resistance. It also included ribosome, proteasome, cleaver, endoplasmic reticulum protein processing, endocytosis, DNA replication, oxidative phosphorylation, apoptosis, metabolic pathway, cancer center carbon metabolism and FOXO signaling pathway as shown in Figure 7. The above data indicated that lncRNA−*MSTRG.16919.1* participated in a variety of biological processes during MDBK infection with BHV−1 virus, thus regulating cells.

### 3.7. Western Blot Verification of KEGG Enrichment in Partial Signal Pathways

According to the results of KEGG enrichment analysis, the top 20 signal pathways were analyzed, the signal pathways related to virus infection were screened out and NF−κB protein, IκB protein and JNK protein in the NF−κB and MAPK signal pathways were detected. After silencing lncRNA−*MSTRG.16919.1*, compared with siRNA−*MSTRG.16919.1NC* group, BHV−33 h group and MDBK group, the expression of NF−κB protein and JNK protein was downregulated and the expression of IKB protein was upregulated, which indicated that silencing lncRNA−*MSTRG.16919.1* affected the two signal pathways of NF−κB and MAPK, as seen in Figure 8.

### 3.8. Protein Interaction Network Analysis Results

Through the STRING protein interaction database (STRING. Available online: http://string−db.org/ (accessed on 12 June 2022)) [19], the differential genes produced by sequencing siRNA−*MSTRG.16919.1* and siRNA−*MSTRG.16919.1NC* groups and siRNA−*MSTRG.16919.1* and BHV−33 h groups were screened, and 498 repeated genes were obtained, all of which met the conditions of difference multiple exceeding one and *p* value less than 0.05. The proteins encoded by 498 differential genes were mapped to form a protein interaction network, as shown in Figure 9. The key genes from the protein interaction data obtained from string by Degree algorithm in Cytoscape software were screened and the top 20 genes obtained by combined score, including BIRC6, PTEN, PIK3CA, RANBP2, SCAF11, TRIP12, STAG2, BTAF1, ATAD2B, PDS5A, USP34, SKIV2L2, LTN1, CLTC, CEP350, KDM6A, JAK2, DMXL1, PIK3CB and MAPK8, among which SKIV2L2, JAK2, PIK3CB and MAPK8 were related to virus infection. This played a role in the process of BHV−1 infecting MDBK cells (Figure 10).

## 4. Discussion

Bovine herpes virus type I (BHV−1) is a member of the alpha herpesvirus subfamily, which can cause many diseases in bovines by proliferating in MDBK cells and causing cytopathic changes. The replication and growth characteristics of BHV−1 have been understood, but the pathogenesis of BHV−1 and the interaction between virus and host needed to be further studied to assist the treatment and prevention of bovine infectious rhinotracheitis.

As a multifunctional regulator of transcription, lncRNA not only participates in the epigenetic control of transcription initiation to the regulation of mature transcription stability, but also participates in many life activities, such as innate immune response and host–virus interaction regulation. Virus infection can cause differential expression of lncRNA in cells and differentially expressed lncRNA can regulate innate immune response in many ways, resisting virus infection. Studies have shown that in A549 cells infected with influenza A virus (IAV), lncRNA−IVRIPE, which inhibits IAV by promoting interferon, and interferon−stimulating gene expression had a significantly increasing trend, while overexpression of IVRIPE inhibits IAV replication [23] and IVRIPE is an important regulator of host antiviral response. Some studies have shown that viruses can negatively regulate the host’s antiviral immune response by using lncRNA encoded by themselves and their hosts, creating an environment for their own replication, so 3’−untranslated region (3’ UTR) of flavivirus genome can transcribe small ncRNA, or flavivirus subgenomic RNA (sfRNA). The lncRNA protects viral RNA from host exoribonuclease 1 (XRN1) degradation in infected cells [24].

The lncRNA−*MSTRG.16919.1* is a novel lncRNA that was significantly increased in BHV−1 infected MDBK cells by whole−transcription sequencing and analysis, but its function remained unclear. There is no research reported on the analysis of mRNA associated with lncRNA−*MSTRG.16919.1* in MDBK cells infected by BHV−1, because its genome is too large to be overexpressed. The lncRNA−*MSTRG.16919.1* gene was therefore silenced and transcriptome sequenced to explore the role of lncRNA−*MSTRG.16919.1* in the process of BHV−1 infecting MDBK cells. Since it was uncertain whether lncRNA−*MSTRG.16919.1* was in the nucleus or cytoplasm, the techniques of RNA interference (RNAi) and (antisense oligonucleotide ASO) were used to silence them as RNAi targets lncRNA in the cytoplasm while ASO targets lncRNA in the nucleus. The sequencing data were analyzed by EdgeR software, and the differential genes generated by the siRNA−*MSTRG.16919.1* and siRNA−*MSTRG.16919.1 NC* group and the siRNA−*MSTRG.16919.1* and BHV−1 33 h group were screened, and a volcano plot and 498 repetitive mRNA genes were obtained. Six differentially expressed mRNA were then selected from repeated genes including three upregulated and three downregulated genes, which were verified by RT−qPCR. The change trend of gene expression was consistent with the sequencing results, which indicated that the accuracy of this sequencing result could be used for subsequent experimental study of downstream functions.

To determine the biological processes of mRNA associated with lncRNA−*MSTRG.16919.1*, GO and KEGG enrichment analyses were conducted on mRNA associated with lncRNA−*MSTRG.16919.1*. The results of GO enrichment analysis showed that the biological processes of mRNA associated with lncRNA−*MSTRG.16919.1* were molecular functions including nucleotide, purine nucleoside and enzyme binding, which indicated that lncRNA−*MSTRG.16919.1* had the above functions. In cellular components, it involved most of the intracellular and cytoskeleton parts, organelles, microtubule cytoskeleton, microtubule tissue center and spindle microtubules, which indicated that this lncRNA participated in the synthesis of cell components. In biological processes, it included cell cycle and glial macromolecule metabolism processes, which showed that lncRNA−*MSTRG.16919.1* was mostly involved in the above biological processes.

The results of KEGG enrichment analysis showed that lncRNA−*MSTRG.16919.1* was involved in cancer, viral infection, cell cycle, apoptosis and other pathways. Among them, Toll and Imd signaling, TNF signaling, NF−κB signaling and MAPK signaling pathways were related to virus infection. The results of GO and KEGG enrichment analyses suggested that lncRNA−*MSTRG.16919.1* was a multifunctional lncRNA and played an important regulatory role in a variety of biological processes. The NF−κB, IκB and JNK proteins were detected in the NF−κB signaling and MAPK signaling pathways by Western blot. The results showed that the expression of NF−κB protein and JNK protein was downregulated, while the expression of IκB protein was increased in the siRNA−*MSTRG.16919.1* group compared with the siRNA−*MSTRG.16919.1NC* group, BHV−33 h group, and MDBK group. The above data indicated that lncRNA−*MSTRG.16919.1* may participate in this signaling pathway and then participate in the regulation of virus infection.

Protein interaction networks encoded by differential genes were constructed using the STRING protein interaction database and the top 20 proteins including RANBP2, KDM6A, ATAD2, PIK3CA, STAG2, DMXL1, CE3P50, BTAF1, PTEN, SKIV2L2, LTN1, BBIRC6, USP34, CLTC, SCAF11, PIK3CB, MAPK8, TRIP12, PDS5A and JAK2 were further screened by Cytoscape. These results confirmed that lncRNA−*MSTRG.16919.1* was a multifunctional lncRNA, which participated in apoptosis by BIRC6, pyroptosis by SCAF11, tumor suppressor and metabolic regulator using PTEN, formation of nuclear pore complexes using RANBP2, viral infection and cellular immunity using SKIV2L2, JAK2, PIK3CB and MAPK8. It has been found that in SKIV2L−deficient cells, the unfolded protein response (UPR) produced an endogenous RLR ligand through the IRE−1 endonuclease lysis of cell RNA, which triggered the production of type I interferon (IFN) and that SKIV2L RNA exocrine can effectively limit the activation of RLRs. Studies have shown that JAK2−V617F promoted the synthesis of PD−L1 in MPN cells, which led to the limitation of the antitumor T cell response and the change of T cell metabolism and finally led to JAK2−V617F driving the immune escape of MPN cells [25].

The JAK2 and IL6/JAK2/STAT3 signaling pathways are therapeutic targets for treating excessive inflammatory response to virus infection, where PIK3CB, the target of miRNA−34a participated in TCR−mediated NF−κB signaling after binding to B7 ligand on antigen presenting cells (APC). This molecular interaction activated the PIK3 complex, triggered phosphorylation of PRKCQ through pyruvate dehydrogenase kinase 1 and led to downstream activation of NF−κB [26].

The MAPK8 gene is also known as JNK and has been identified as being affected by viral infection. Using an overexpression experiment, Fung showed that JNK/MAPK8 participated in coronavirus–host interaction [27], so silencing lncRNA*−MSTRG.16919.1* may affect the expression of SKIV2L2, JAK2, PIK3CB and MAPK8 proteins and then regulate the infection of MDBK cells by BHV−1.

## 5. Conclusions

In this study, lncRNA−*MSTRG.16919.1* was silenced for the first time to study the role of lncRNA−*MSTRG.16919.1* in the process of BHV−1 infecting MDBK cells. Based on the above data and analysis, we can draw the following conclusions. (1) lncRNA−*MSTRG.16919.1* is a multifunctional lncRNA and plays a significant role in cell cycle, cell metabolism, and virus infection during BHV−1 infection of MDBK cells. (2) lncRNA−*MSTRG.16919.1* may respond to BHV−1 infection through NF−κB signaling pathway and MAPK signaling pathway. (3) lncRNA−*MSTRG.16919.1* may regulate the infection of MDBK cells by BHV−1 by affecting the expression of SKIV2L2, JAK2, PIK3CB and MAPK8 proteins.

These results elucidate the possible mechanism of lncRNA−*MSTRG.16919.1* in the process of BHV−1 infecting MDBK cells.

## Figures and Tables

**Figure 1 viruses-14-02104-f001:**
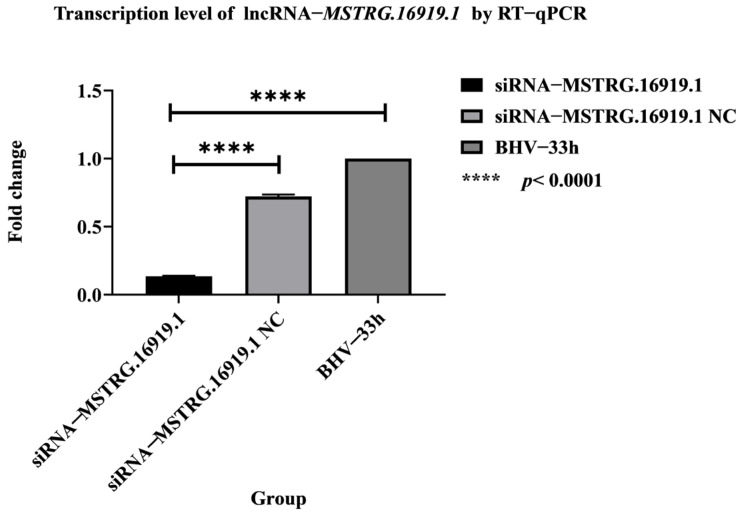
lncRNA−*MSTRG.16919.1* silencing effect detection. siRNA−MSTRG.16919.1 is the experimental group in which MDBK cells were infected with BHV−1 for 33 h and lncRNA−*MSTRG.16919.1* silenced; siRNA−MSTRG.16919.1 MDBK cells infected with BHV−1 for 33 h and silencing an unrelated sequence is the negative control group of the siRNA−*MSTRG.16919.1* group; the BHV−1 33 h group is MDBK cells infected with BHV−1 for 33 h (reference group).

**Figure 2 viruses-14-02104-f002:**
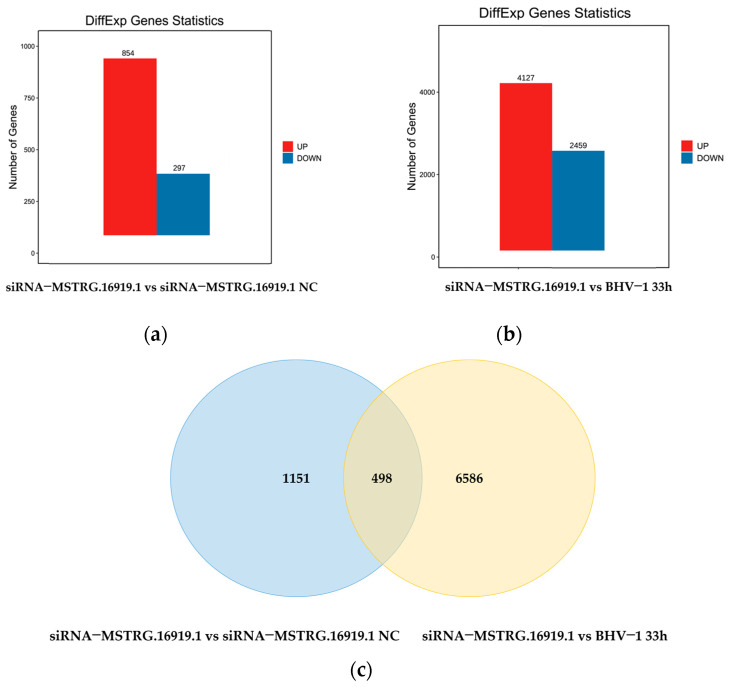
Statistical diagram of differential genes. (**a**) Gene statistics of the difference between siRNA−*MSTRG.16919.1* and siRNA−*MSTRG.16919.1 NC*; (**b**) siRNA−*MSTRG.16919.*1 and BHV−1 33 h differential genes statistics; (**c**) repeat gene screening for groups a and b.

**Figure 3 viruses-14-02104-f003:**
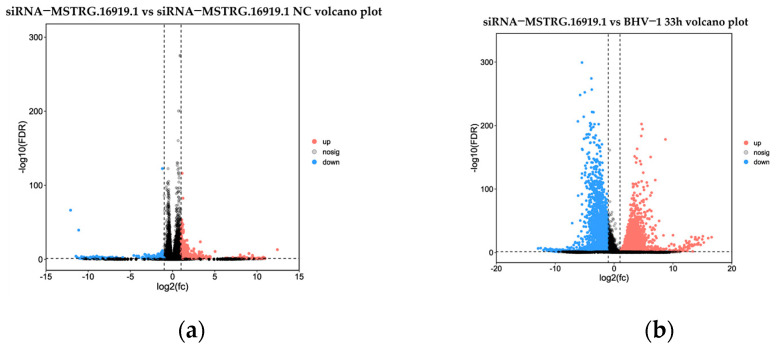
Volcanic map of difference comparisons. (**a**) siRNA−*MSTRG.16919.1* and siRNA−*MSTRG.16919.1 NC* volcano plot; (**b**) siRNA−*MSTRG.6919.*1 and BHV−1 33 h volcano plot. Red dots indicate genes with upregulated transcription level, blue dots indicate genes with downregulated transcription level, and black dots indicate genes whose transcription levels have not changed differently.

**Figure 4 viruses-14-02104-f004:**
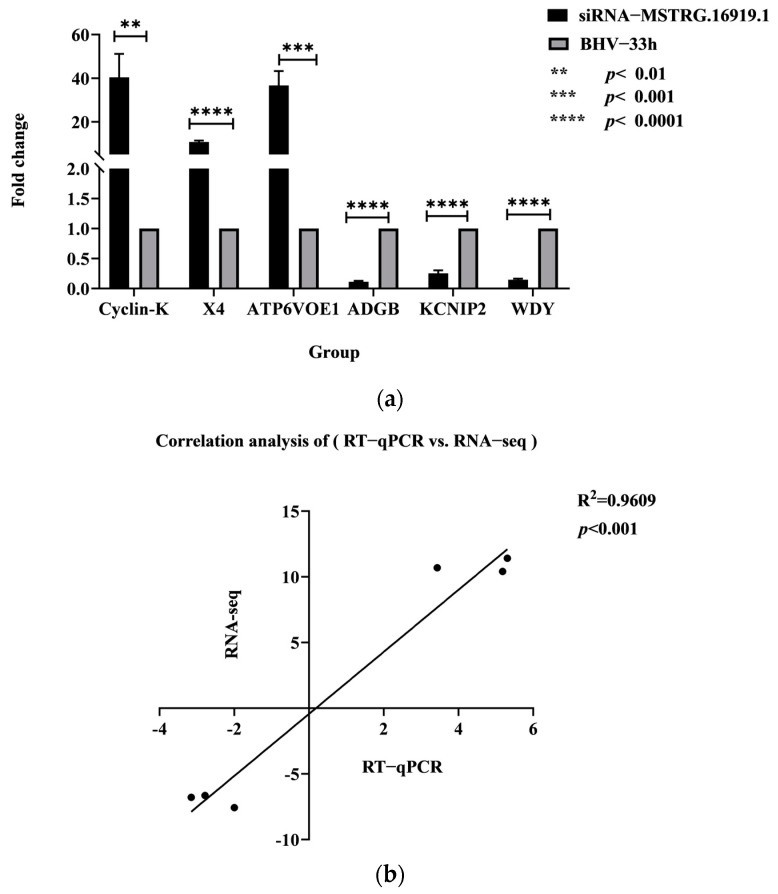
RT−qPCR verification results. (**a**) The abscissa denotes the name of the genes and the ordinate denotes the multiple of difference. The first three genes are upregulated and the last three genes are downregulated. (**b**) Correlation analysis of (RT−qPCR vs. RNA−seq).

**Figure 5 viruses-14-02104-f005:**
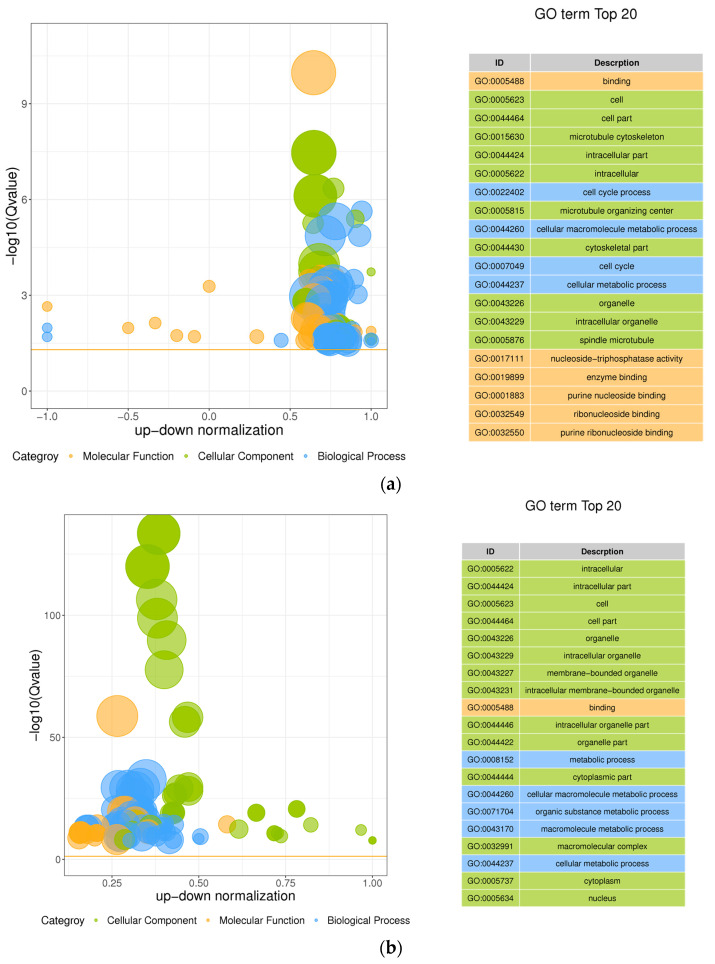
The GO enrichment bubble diagram of difference. (**a**) siRNA−*MSTRG.16919.1* and siRNA−*MSTRG.16919.1NC* bubble chart; (**b**) siRNA−*MSTRG.16919.1* and BHV−1 33 h bubble chart. The ordinate is log10 (Q value), the abscissa is the z−score (the ratio of the difference between the number of upregulated differential genes and the number of downregulated differential genes to the total differential genes), and the yellow line represents the threshold of Q value = 0.05. On the right is a list of the top 20 GO terms of Q values. Assorted colors represent different ontology, in which green represents molecular function, yellow represents cellular component, and blue represents biological process. The bubble size represents the number.

**Figure 6 viruses-14-02104-f006:**
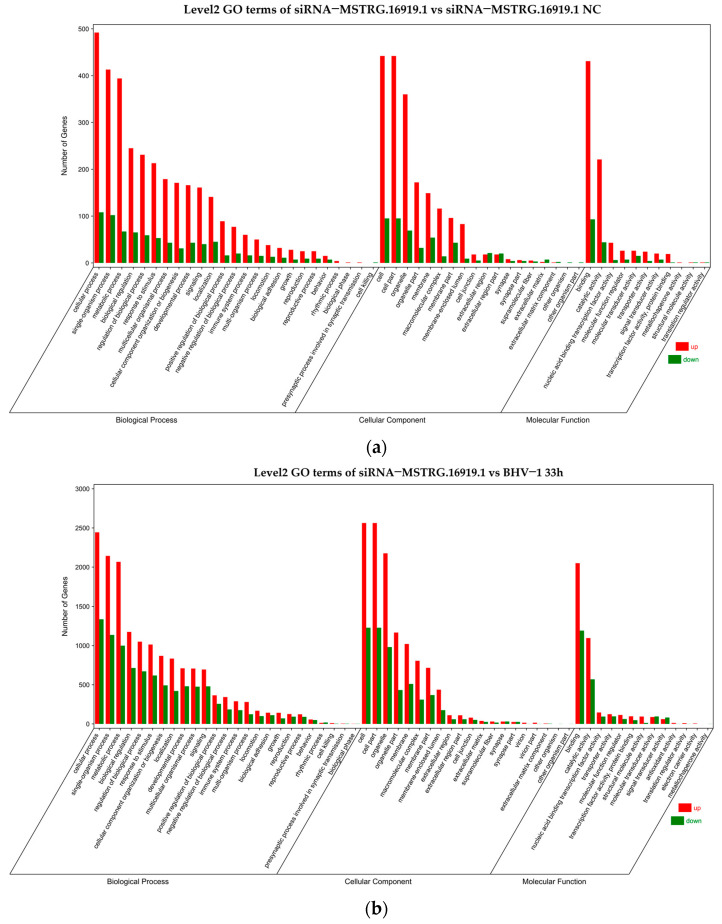
GO enrichment classification histogram. (**a**) Level 2 GO terms of siRNA−*MSTRG.16919.1* and siRNA−*MSTRG.16919.1 NC*; (**b**) level 2 GO terms of siRNA−*MSTRG.16919.1* and BHV−1 33 h. The abscissa is the second−level GO term, and the ordinate is the number of differential genes in the term. Red indicates upregulation and green indicates downregulation.

**Figure 7 viruses-14-02104-f007:**
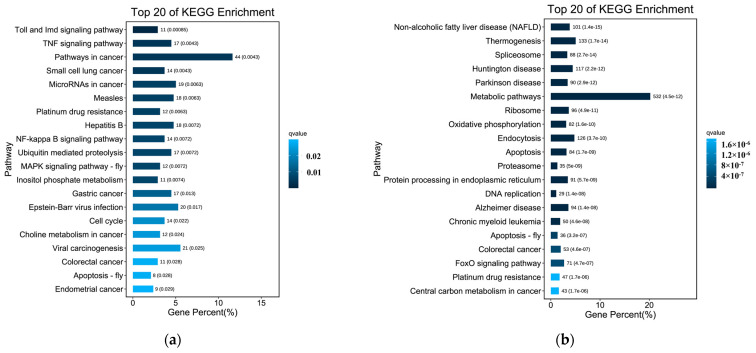
KEGG enrichment bar chart. (**a**) KEGG enrichment bar chart for siRNA−*MSTRG.16919.1* and siRNA−*MSTRG.16919.1 NC*; (**b**) KEGG enrichment bar chart for siRNA−*MSTRG.16919.1* and BHV−33 h. The ordinate is path and the abscissa is the percentage of the number of pathways to the number of all differential genes. The darker the color is, the smaller the Q value is. The values on the column are the number of pathways and Q value.

**Figure 8 viruses-14-02104-f008:**
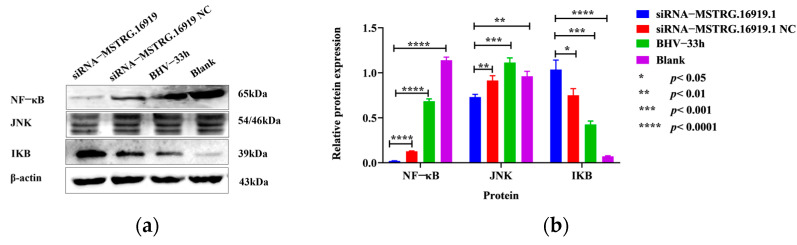
Western blot results. siRNA−*MSTRG.16919.1* group is the experimental group in which MDBK cells were infected with BHV−1 for 33 h and silenced lncRNA−*MSTRG.16919.1*; siRNA−*MSTRG.16919.1NC* group was MDBK cells infected with BHV−1 for 33 h and silenced an unrelated sequence, is the negative control group of siRNA−*MSTRG.16919.1* group; the BHV−1 33 h group is the MDBK cell infected with BHV−1 33 h is the reference group; the blank group is the MDBK cell group without any treatment; (**a**) Western blot results; (**b**) Quantitative analysis of Figure 8a.

**Figure 9 viruses-14-02104-f009:**
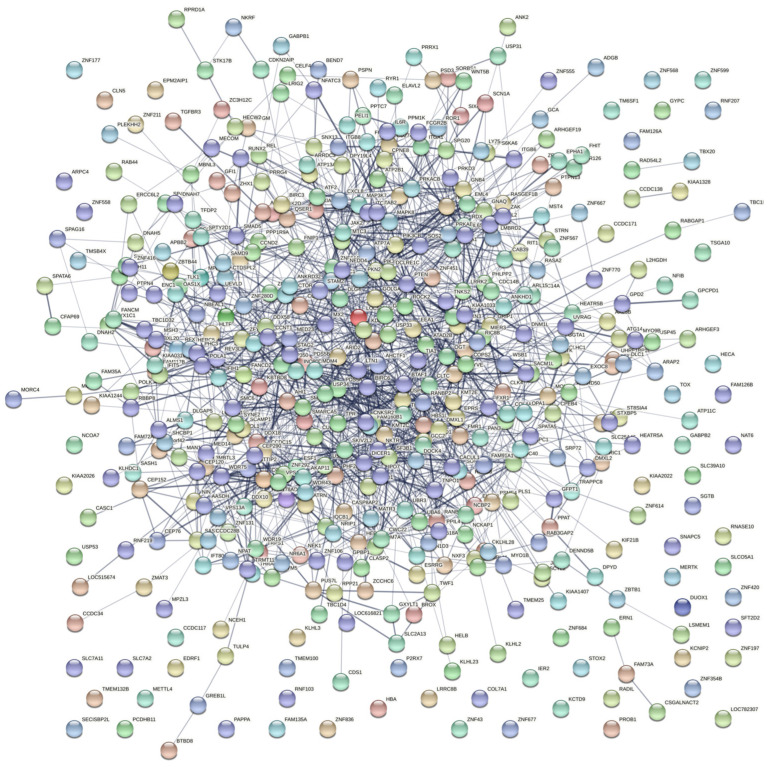
Protein interaction network diagram. Network nodes represent proteins: splice isoforms or posttranslational modifications are collapsed, i.e., each node represents all the proteins produced by a single, protein−coding gene locus; node color: colored nodes, query proteins and first shell of interactors; white nodes, second shell of interactors; node content: empty nodes, proteins of unknown 3D structure; filled nodes, some 3D structure is known or predicted; edges represent protein–protein associations; associations are meant to be specific and meaningful, i.e., proteins jointly contribute to a shared function, but this does not necessarily mean they are physically binding to each other.

**Figure 10 viruses-14-02104-f010:**
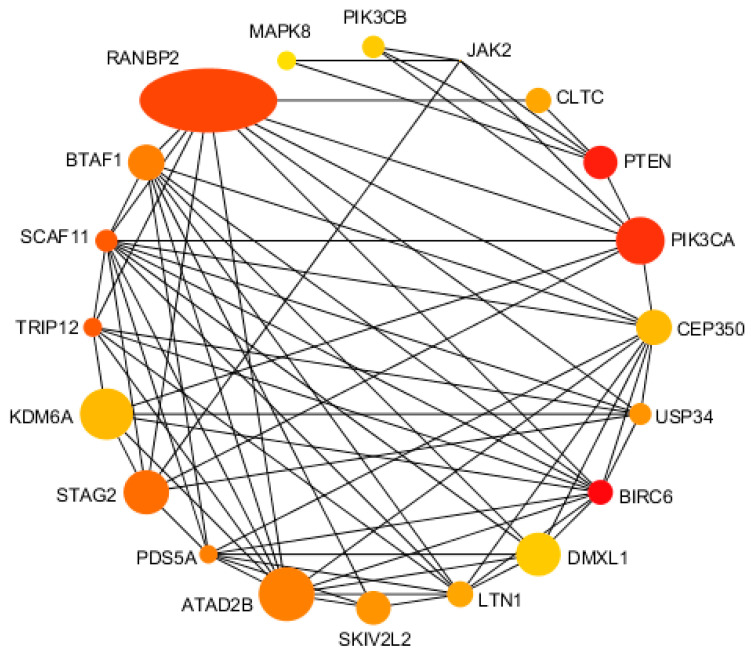
Top 20 proteins. The size of the dot represents the difference multiple, and the larger the difference multiple, the larger the dot, the smaller the difference multiple and the smaller the dot. The dot color represents the score of Degree algorithm. The redder the color, the higher the score, and the yellower the score. The connection between dots indicates the interaction between proteins.

**Table 1 viruses-14-02104-t001:** The primer sequence of RT−qPCR.

Gene	Forward Primer	Reverse Primer	Product Length (bp)
Cyclin−K	AATAAGGACAGAACCAGGCACAAGG	TCGGGAGAGGCTTCCAGTAAACC	143
X4	GGAGCTGTAAATCCTGGAAGCC	TCCCTGATGTGTGGTTCTGGC	114
ATP6VOE1	CCAGTGCCCAGTGATTGAGGTG	GTCAAAGAAGGTGGTCTGGCTATGG	80
ADGB	GCTGGTTCCAGGATCTGGTT	TGCTAGTTCATCTTTACCTTTTCCG	151
KCNIP2	TCGAGCACATACGCCACTTT	AGGTTGAAGGCCCAGTTCAG	140
WDY	AGTTCATCGCCACCTCCTCCTAC	ACACACAGTCGTTCACATCCTTCAC	141
UCHL5	ACAAAGACAACTTGCTGAGGAACCC	GGCAACCTCTGACTGAATAGCACTT	79

**Table 2 viruses-14-02104-t002:** Statistics of data filtering.

Sample	Raw Data	Clean Data (%)	Adapter (%)	Low Quality (%)	PolyA (%)	N (%)
siRNA−*MSTRG.16919.1*	49,105,820	48,782,776 (99.34%)	22,812 (0.05%)	300,232 (0.61%)	0 (0.00%)	0 (0.00%)
siRNA−*MSTRG.16919.1NC*	55,783,530	55,369,610 (99.26%)	32,178 (0.06%)	381,742 (0.68%)	0 (0.00%)	0 (0.00%)
BHV−33 h	1,090,296,987	108,777,966 (99.77%)	47,012 (0.04%)	202,008 (0.19)	0 (0.00%)	2712 (0.00%)

## Data Availability

The findings of this study are available within this paper.

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
