# Peer review of "Analysis of the Function of LncRNA-MSTRG.16919.1 in BHV-1-Infected Bovine Kidney Subculture Cells by Transcriptome Sequencing"

_viruses, 2022, doi:10.3390/v14102104_

Round 1
Reviewer 1 Report
In this paper, the authors analyse the consequences of reduction of levels of a long non-coding RNA (LncRNA-MSTRG.16919.1) in MDBK cells compared with BHV-1 infection of the cells and in cells transfected by a control siRNA. The analysis has been done by RNA-Seq and the methods and statistical packages seem appropriate. Inevitable very large amounts of data have been generated and, at the end of this work, it's not clear (at least to this reader) how this throws light on BHV-1 pathogenesis.
Major points
1. It's not clear what level of reduction of the Lnc RNA was achieved by siRNA treatment. One would have thought that this would be Figure 1. So it's difficult to judge the significance of the work as we don't know if there was a 90% or a 50% reduction. This data ought to be supplied.
2. In Fig2, the colour of the dots in the graph does not correspond to the description in the Figure legend.
3. In Fig 3, it's not clear what Blank means in the chart - does it mean control siRNA-treated cells? A similar question applies to Fig 7.
4. The character size in Figs 4 and 5 is much too small to read, rendering these Figures of very limited value. Please increase the size!
5. Fig 7 has no legend.
6. Again Fig 8 is impossible to read
Minor points
1. The text around line 199, the description of the Western blot, is difficult to understand. It's not clear what is meant. Also in lines 2017 and 208 it's not clear what "gap-associated protein gap" means.
Author Response
Dear Editors and Reviewers:
Thank you for your letter and for the reviewers’ comments concerning our manuscript entitled “The analysis of the function of LncRNA-MSTRG.16919.1 in BHV-1-infected bovine kidney subculture cells by transcriptome sequencing”(ID: viruses-1876063). Those comments are all valuable and very helpful for revising and improving our papers, as well as the important guiding significance to our researches. We have studied comments carefully and have made correction which we hope meet with approval. Revised portion are marked in red in the paper. The main corrections in the paper and responds to the reviewer’s comments are as flowing:
Responds to the reviewer’s comments:
Major points
- Comment:“It's not clear what level of reduction of the LncRNA was achieved by siRNA treatment. One would have thought that this would be Figure 1. So it's difficult to judge the significance of the work as we don't know if there was a 90% or a 50% reduction. This data ought to be supplied”.
Response: We thank the reviewer for raising this question.
We have added the RT-qPCR validation data of the lncRNA-MSTRG.16919.1 silencing effect. On pages 5 to 6 of the article, there is the 3.1 section of the result, which is Figure 1 of the revised article.
Figure 1 in the original article is a revised Figure 2, which represents the differential gene changes obtained after sequencing. The left picture is the comparison of siRNA-MSTRG.16919.1 and siRNA-MSTRG.16919.1 NC to obtain the differential gene situation; the right picture is the comparison of siRNA-MSTRG.16919.1 and BHV-33h to obtain the differential gene situation.
- Comment:“In Fig2, the colour of the dots in the graph does not correspond to the description in the Figure legend”.
Response: We are so grateful for your kind question.
In the figure, the dots representing down-regulated genes are in blue, which has been modified in the legend, and the modified position is on page 8, line 267.
- Comment:“In Fig 3, it's not clear what Blank means in the chart - does it mean control siRNA-treated cells? A similar question applies to Fig 7.”
Response: We thank the reviewer for raising this question.
I have redefined each grouping. The experimental groups are as follows: siRNA-MSTRG.16919.1 group is the experimental group in which MDBK cells were infected with BHV-1 for 33h and silenced lncRNA-MSTRG.16919.1; siRNA-MSTRG.16919.1 NC group was MDBK cells infected with BHV-1 for 33h and silenced an unrelated sequence, is the negative control group of the siRNA-MSTRG.16919.1 group; the BHV-1 33h group is the MDBK cell infected with BHV-1 33h as the reference group; the blank group is the MDBK cell group without any treatment.
The original Figure 3 is the current Figure 4(a), which has been modified and the groupings have been marked in the legend. In order to more comprehensively illustrate our verification results, the RT-qPCR and RNA-seq correlation curve analysis is added as Figure 4(b) in the current article.
Figure 7 in the original article is Figure 8 in the current article, and the grouping information is added in both the figure and the legend.
- Comment:“The character size in Figs 4 and 5 is much too small to read, rendering these Figures of very limited value. Please increase the size!”.
Response: We are so grateful for your kind question.
The picture has been enlarged, and it is Figure 5 and Figure 6 in the current article.
- Comment:“Fig 7 has no legend”.
Response: We thank the reviewer for raising this question.
Figure 7 in the original article is Figure 8 in the current article, and the grouping information is added in both the figure and the legend.
- Comment:“Again Fig 8 is impossible to read”.
Response: We are so grateful for your kind question.
Legend has been added, on page 13-14, lines 371-378.
Minor points
- Comment:“The text around line 199, the description of the Western blot, is difficult to understand. It's not clear what is meant. Also in lines 2017 and 208 it's not clear what "gap-associated protein gap" means”.
Response: We thank the reviewer for raising this question.
Western blot description modified to read “Skimmed milk was removed, and the NC membrane was washed with TBST three times”. The location is on page 5, line 202 of the current article.
The content description of "gap-associated protein gap" in lines 207-208 of the original article is inaccurate and has been deleted.
We appreciate for Editors/Reviewers’ warm work earnestly, and hope that the correction will meet with approval. We are looking forward to hearing from you soon.
With best wishes,
Sincerely,
Fan Zhang

Reviewer 2 Report
I want to congratulate the authors for the work that scientifically sound. I have no particular comments to make.
Author Response
Dear Editors and Reviewers:
Thank you for your letter and for the reviewers’ comments concerning our manuscript entitled “The analysis of the function of LncRNA-MSTRG.16919.1 in BHV-1-infected bovine kidney subculture cells by transcriptome sequencing”(ID: viruses-1876063).Those comments are all valuable and very helpful for revising and improving our papers, as well as the important guiding significance to our researches. We have studied comments carefully and have made correction which we hope meet with approval. Revised portion are marked in red in the paper.
We appreciate for Editors/Reviewers’ warm work earnestly, and hope that the correction will meet with approval. We are looking forward to hearing from you soon.
With best wishes,
Sincerely,
Fan Zhang

Reviewer 3 Report
Dear Editor,
The manuscript entitled “The analysis of the function of LncRNA-MSTRG.16919.1 in BHV-1-infected bovine kidney subculture cells by transcriptome sequencing” by Fan Zhang et al. presents a preliminary study on the role of lncRNA-MSTRG.16919.1 in bovine herpes virus type I (BHV-1) infected MDBK cells. The authors have silenced the lncRNA-MSTRG.16919.1 gene and performed transcriptome analysis on cell culture. The sequencing data were analyzed and confirming experiments (RT-PCR and western blots).
Τhe manuscripts’ objects are interesting, the authors have collected and analyze a vast amount of data, however the data presentation is somewhat confusing and data interpretation seems incomplete. Therefore, the manuscript could be accepted for publication after major revisions. My detailed comments for the authors to consider are provided below:
1. In my opinion the abstract should be re-written: a brief comment on virus importance should be included at the beginning of the abstract and the rest should be re-phrased because at its present form reminds a list of results
2. Page 2, lines 57-62: More information on lncRNA-MSTRG.16919.1 should be provided. Reference [12] is in Chinese, therefore I cannot read it (and a great portion of Viruses reader I suppose). The authors should explain how they chose that specific lnc, its characteristics, under what experimental conditions it was obtained, the methodology used, etc.
3. Page 2, lines 81-85: It is not clear what each sample is, i.e. lncRNA-MSTRG.16919.1 represents the silenced sample and lncRNA-MSTRG.16919.1NC is the ‘wild type’/ normal sample? Is MDBK infected BHV-1 33 hours group (BHV-33h) a normal sample in different time point and how/ why was chosen as a reference group? Also, the sentence on lines 83-85 is not clear, please explain.
4. Each referred sample was consisting only by one biological replicate?
5. Why ubiquitic C-terminal hydrolase L5 was used as reference gene? Please provide relative published references
6. Page 4, lines 156-162, 174-179: This general information are not needed in the M&M section.
7. Please provide some results/comments on your specific silencing experiments.
8. Table 2 column titles are confusing: what is Laudatas and clean grand view?
9. Paragraphs 3.2 and 3.3. contain overlapping information and can be merged. Figures 1 and 2 can be combined in one figure.
10. Figure 3 needs to contain comparison of RNA-seq data with RT-PCR results, not RT-PCR results alone.
11. In figure 4, what the bubble size represents?
12. Page 9, lines 304-308: Please re-phrase, this sentence is very confusing.
13. Figure 7. Please explain the samples coding. What is each depicted sample?
14. Page 13, line 388: Which are the two groups? Please explain more comprehensively.
15. Are there any other studies on bovine viruses and lncRNAs?
16. Based on all data and analysis, please provide a possible function for the studied lncRNA, as a conclusion.
Author Response
Dear Editors and Reviewers:
Thank you for your letter and for the reviewers’ comments concerning our manuscript entitled “The analysis of the function of LncRNA-MSTRG.16919.1 in BHV-1-infected bovine kidney subculture cells by transcriptome sequencing”(ID: viruses-1876063).Those comments are all valuable and very helpful for revising and improving our papers, as well as the important guiding significance to our researches. We have studied comments carefully and have made correction which we hope meet with approval. Revised portion are marked in red in the paper. The main corrections in the paper and responds to the reviewer’s comments are as flowing:
Responds to the reviewer’s comments:
Major points
- Comment:“In my opinion the abstract should be re-written: a brief comment on virus importance should be included at the beginning of the abstract and the rest should be re-phrased because at its present form reminds a list of results”.
Response: We thank the reviewer for raising this question.
The summary has been modified to add virus-related information.
- Comment:“Page 2, lines 57-62: More information on lncRNA-MSTRG.16919.1 should be provided. Reference [12] is in Chinese, therefore I cannot read it (and a great portion of Viruses reader I suppose). The authors should explain how they chose that specific lnc, its characteristics, under what experimental conditions it was obtained, the methodology used, etc.”.
Response: We are so grateful for your kind question.
The literature has been re-linked.
MDBK cells were infected with BHV-1 in the early stage, and RNA samples were collected at 18h, 24h and 33h, respectively, and the whole transcriptome was sequenced. According to the previous sequencing data, we found that the expression of lncRNA-MSTRG.16919. 1 increased significantly after 33 hours of virus infection, so we inferred that the up-regulation of this gene expression might be related to BHV-1 infection. Further analysis of sequencing data showed that the gene associated with lncRNA-MSTRG.16919. 1 was quite correct, so we selected it as the target gene for further study.
As lncRNA-MSTRG.16919. 1 is a newly assembled gene, there is no report yet, and its specific functions and characteristics are still under further study.
- Comment:“Page 2, lines 81-85: It is not clear what each sample is, i.e. lncRNA-MSTRG.16919.1 represents the silenced sample and lncRNA-MSTRG.16919.1NC is the ‘wild type’/ normal sample? Is MDBK infected BHV-1 33 hours group (BHV-33h) a normal sample in different time point and how/ why was chosen as a reference group? Also, the sentence on lines 83-85 is not clear, please explain”.
Response: We thank the reviewer for raising this question.
I have redefined each grouping. The experimental groups are as follows: siRNA-MSTRG.16919.1 group is the experimental group in which MDBK cells were infected with BHV-1 for 33h and silenced lncRNA-MSTRG.16919.1; siRNA-MSTRG.16919.1 NC group was MDBK cells infected with BHV-1 for 33h and silenced an unrelated sequence, is the negative control group of the siRNA-MSTRG.16919.1 group; the BHV-1 33h group is the MDBK cell infected with BHV-1 33h as the reference group; the blank group is the MDBK cell group without any treatment.
MDBK cells were infected with BHV-1 in the early stage, and RNA samples were collected at 18h, 24h and 33h, respectively, and the whole transcriptome was sequenced. We found that lncRNA-MSTRG.16919. 1 expressed most significantly at 33h after virus infection, so BHV-33h was used as a reference for subsequent comparison.
There are errors in sentences 83-85, which have been corrected. On page 2, lines 92-93 of the current article.
- Comment:“Each referred sample was consisting only by one biological replicate?”.
Response: We are so grateful for your kind question.
Each reference sample was repeated at least 3 times.
- Comment:“Why ubiquitic C-terminal hydrolase L5 was used as reference gene? Please provide relative published references”.
Response: We thank the reviewer for raising this question.
In the early stage, we used β-actin as an internal reference when verifying whole transcriptome sequencing. Through RT-qPCR data, we found that the value of β-actin varied from group to group. Then we further analyzed the data of transcriptome sequencing and found the value of β-actin have a slight variation between the different groups. The possible reason is that cells produce obvious cytopathic changes by BHV-1 infected and mRNA of β-actin acts as a protein of the cytoskeleton were degraded. It cannot be used as an internal reference. We finally decided to use UCHL5 as an internal reference through the analysis of transcriptome sequencing data and literature review and the literature has been added to the article [14]. Introduced at lines 155-156 on page 4 of the article.
- Comment:“Page 4, lines 156-162, 174-179: This general information are not needed in the M&M section”.
Response: We are so grateful for your kind question.
This section has been deleted
- Comment:“Please provide some results/comments on your specific silencing experiments.”.
Response: We thank the reviewer for raising this question.
We have added the RT-qPCR validation data of the lncRNA-MSTRG.16919.1 silencing effect. On pages 5 to 6 of the article, there is the 3.1 section of the result, which is Figure 1 of the revised article.
- Comment:“Table 2 column titles are confusing: what is Laudatas and clean grand view?”.
Response: We are so grateful for your kind question.
Table 2 is incorrect and has been revised: Laudatas-Rawdatasï¼›clean grand view-CleanDate.
- Comment:“Paragraphs 3.2 and 3.3. contain overlapping information and can be merged. Figures 1 and 2 can be combined in one figure.”.
Response: We thank the reviewer for raising this question.
Paragraphs 3.2 and 3.3 show the same problem from different aspects. the content has been integrated, but the figure 1 and 2 cannot be combined in one figure. This part of 3.3 in the article results is the integration result.
- Comment:“Figure 3 needs to contain comparison of RNA-seq data with RT-PCR results, not RT-PCR results alone”.
Response: We are so grateful for your kind question.
The RT-qPCR and RNA-seq correlation curve analysis is added as Figure 4(b) in the current article.
- Comment:“In figure 4, what the bubble size represents?”.
Response: We thank the reviewer for raising this question.
The bubble size represents the number.
- Comment:“Page 9, lines 304-308: Please re-phrase, this sentence is very confusing.”.
Response: We are so grateful for your kind question.
This part has been modified. On page 11, lines 318-322.
- Comment:“Figure 7. Please explain the samples coding. What is each depicted sample?”.
Response: We thank the reviewer for raising this question.
Figure 7 in the original article is Figure 8 in the current article, and the grouping information is added in both the figure and the legend.
The experimental groups are as follows: siRNA-MSTRG.16919.1 group is the experimental group in which MDBK cells were infected with BHV-1 for 33h and silenced lncRNA-MSTRG.16919.1; siRNA-MSTRG.16919.1 NC group was MDBK cells infected with BHV-1 for 33h and silenced an unrelated sequence, is the negative control group of the siRNA-MSTRG.16919.1 group; the BHV-1 33h group is the MDBK cell infected with BHV-1 33h as the reference group; the blank group is the MDBK cell group without any treatment.
- Comment:“Page 13, line 388: Which are the two groups? Please explain more comprehensively”.
Response: We are so grateful for your kind question.
This part has been revised, page 15 of the previous article, lines 414-417.
- Comment:“Are there any other studies on bovine viruses and lncRNAs?”.
Response: We thank the reviewer for raising this question.
Reference 11 is the previous research on BHV-1 and lncRNA in our laboratory, which is still being further explored, and no other studies on BHV-1 and lncRNAs have been reported. But there are studies on another bovine virus (for example BVDV) and lncRNAs (no lncRNA-MSTRG.16919.1).
- Comment:“Based on all data and analysis, please provide a possible function for the studied lncRNA, as a conclusion”.
Response: We are so grateful for your kind question.
We have added possible function of LncRNA as a conclusion. In this article, on page 16, lines 472-483.
We appreciate for Editors/Reviewers’ warm work earnestly, and hope that the correction will meet with approval. We are looking forward to hearing from you soon.
With best wishes,
Sincerely,
Fan Zhang

Round 2
Reviewer 1 Report
No further comments